# Land Use and Land Cover Changes and Its Impact on Soil Erosion in Stung Sangkae Catchment of Cambodia

Nareth Nut [1,*], Machito Mihara [2], Jaehak Jeong [3], Bunthan Ngo [1], Gilbert Sigua [4], P.V. Vara Prasad [5] and Manny R. Reyes [5]

1   Faculty of Agricultural Engineering, Royal University of Agriculture, Phnom Penh 12400, Cambodia; nbunthan@rua.edu.kh
2   Faculty of Regional Environment Science, Tokyo University of Agriculture, Tokyo 156-8502, Japan; m-mihara@nodai.ac.jp
3   Blackland Research & Extension Center, 720 E. Blackland Rd., Temple, TX 76520, USA; jjeong@brc.tamus.edu
4   Agriculture Research Service, Coastal Plains Research Center, United States Department of Agriculture, 2611 West Lucas Street, Florence, SC 29501, USA; gilbert.Sigua@ars.usda.gov
5   Sustainable Intensification Innovation Lab (SIIL), Kansas State University, Manhattan, KS 66506, USA; vara@ksu.edu (P.V.V.P.); mannyreyes@ksu.edu (M.R.R.)
*   Correspondence: nnareth@rua.edu.kh; Tel.: +855-89-222-402 or +855-98-404-130

**Abstract:** Agricultural expansion and urban development without proper soil erosion control measures have become major environmental problems in Cambodia. Due to a high population growth rate and increased economic activities, land use and land cover (LULC) changes will cause environmental disturbances, particularly soil erosion. This research aimed to estimate total amounts of soil loss using the Revised Universal Soil Loss Equation (RUSLE) model within a Geographic Information System (GIS) environment. LULC maps of Japan International Cooperation Agency (JICA) 2002 and Mekong River Commission (MRC) 2015 were used to evaluate the impact of LULC on soil erosion loss in Stung Sangkae catchment. LULC dynamics for the study periods in Stung Sangkae catchment showed that the catchment experienced a rapid conversion of forests to paddy rice fields and other croplands. The results indicated that the average soil loss from the catchment was 3.1 and 7.6 t/ha/y for the 2002 and 2015 periods, respectively. The estimated total soil loss in the 2002 and 2015 periods was 1.9 million t/y and 4.5 million t/y, respectively. The soil erosion was accelerated by steep slopes combined with the high velocity and erosivity of stormwater runoff. The spatial distribution of soil loss showed that the highest value (14.3 to 62.9 t/ha/y) was recorded in the central, southwestern and upland parts of the catchment. It is recommended that priority should be given to erosion hot spot areas, and appropriate soil and water conservation practices should be adopted to restore degraded lands.

**Keywords:** soil erosion; RUSLE; GIS; LULC; Stung Sangkae catchment; Cambodia

## 1. Introduction

Soil erosion is a major environmental and economic concern in most parts of the world [1–4] and poses a threat to land, freshwater, and oceans [5]. This threat may result in low agricultural productivity [6], ecological degradation and high sedimentation [3,7,8]. Almost 84% of global soil loss results from soil erosion processes [9]. According to Ashiagbor et al. [10], and Marondedze et al. [9], the estimated average soil erosion rate across the globe ranges between 12 and 15 t/ha/y.

It is estimated that the average soil erosion by water exceeds 2000 t/km$^2$/y, which mainly occurs on croplands in tropical areas [3,11]. It is reported that soil erosion by human activities is 10–15 times higher than any natural process [12]. For instance, approximately 80% of cultivated areas worldwide face high to severe erosion. The amount of generated sediments can increase the turbidity of waterways and further raise the concentration of

impurities [13]. Furthermore, soil erosion and sediment yield can severely affect people and the environment if the quantity of the sediment exceeds the value of the typical measurement of aquatic organisms. Soil erosion is the central part of the early development of sediment conveyance to streams. In this early development process, evacuated soil particles are converted into sediments due to the effect of the erosion agent [3]. Rainfall, topography, soil characteristics, vegetation or land cover changes, cropping systems and land management practices are the main fundamental factors causing the rate and severity of soil erosion [14,15].

In the past, soil erosion studies were done through physical field assessments [6]. These were more challenging, costly and unfeasible to do mapping of soil erosion risks in huge spatial areas with complex environments in most cases [15,16]. However, even though it is challenging, field-based assessments are needed to provide accurate and reliable data, which are important for the calibration and validation of results from soil loss models [6,17,18]. In previous times, the mapping of soil loss and soil erosion risk have been assessed using altered empirical and stochastic models at local and global levels [19]. A review study by Benavidez et al. [4] summarized 35 previous studies that have applied the USLE (Universal Soil Loss Equation) and RUSLE (Revised Universal Soil Loss Equation) from 1988 to 2017. The review identified modeling techniques developed and used to study soil loss from a field, a hill slope or a catchment/watershed and discussed the different sub-factors of USLE and RUSLE, and analyzed how various studies around the globe have adapted the equations to local environments [4]. These models have different geomorphological parameters that vary in extent and period of application, manipulating factors, processes, features examined, algorithms used and type of assessment. Among the models, RUSLE by Renard [20], which is used to estimate sheet and rill erosion of annual soil loss per unit land area, has arisen as the most widely and globally used model [21].

Some recent studies that have applied the RUSLE model to investigate the soil loss from the land use and land cover (LULC) changes include those by Gashaw et al. [22]; Mustefa et al. [23]; Kidane et al. [24]; Gelagay and Minale [25]; Tadesse et al. [26]; Balabathina et al. [27]; Ayele et al. [28] in Ethiopia; Kogo et al. [15] in Kenya; Marond-edze and Schütt [9] in Zimbabwe; Hui et al. [29] in China; Prasannakumar et al. [30]; Javed et al. [31]; Kolli et al. [32] in India; Talchabhadel et al. [33]; Koirala et al. [34] in Nepal; Thuy and Lee [35] in Vietnam; Plangoen et al. [36] in Thailand; and Chuenchum et al. [3,37] in Mekong River Basin.

There are a wide range of empirical, conceptual, and physical-based models which have been developed to estimate soil loss risks. These models vary in complexity, data requirements, consideration processes, and calibration [9,38,39].

Empirical models such as USLE [40], MUSLE [41] and RUSLE [20] are primarily based on observed data and the relationships between different factors and soil erosion levels. The empirical models require relatively fewer input data compared to conceptual or physical-based models. Thus, empirical models are often used when there is a limitation of data availability. Most empirical models do not provide information about deposition of stream sedimentation which limits their application in modeling mass balance [39]. According to Stefanidis et al. [42] the most commonly used empirical erosion model is USLE [40], and the revised version RUSLE [20]. The main advantages of the USLE/RUSLE model are flexibility, data availability and extensive literature research, making this method suitable for almost all types of conditions and environments [43].

Conceptual models such as Agricultural Policy/Environmental eXtender (APEX) [44] and Soil and Water Assessment Tool (SWAT) [45], are primarily based on sediment and runoff continuity equations, and essentially are hybrids of physical-based and empirical models [46]. Most of the conceptual models use equations from empirical approaches. For instance, empirical model like USLE and MUSLE are carried out in APEX and SWAT to estimate soil erosion.

Physical-based models such as Environmental Policy Integrated Climate (EPIC) [47], APEX [44] and Water Erosion Prediction Project (WEPP) [48], are more capable of re-

sponding to event-based or continuous storms to simulate surface runoff, soil detachment, transportation, and sediment yield [39]. For example, the EPIC model considers the effect of several best management practices (BMPs) related to crop, soil, and nutrient management on soil erosion and soil productivity.

Reyes et al. [49] evaluated erosion prediction of the GLEAMS, RUSLE, EPIC, WEPP (GREW) models to predict the soil loss in various tillage systems. The preliminary results showed that none of the GREW models predicted satisfactorily the soil loss. However, GREW's poor performance in soil loss prediction may not be due to weaknesses in these models' simulation routines. This may result from not choosing the right or appropriate values for some parameters, as well as the observed database of monitoring was also short (only 17 months).

Among the other models, SWAT [45], USLE/RUSLE, APEX [44] or WEPP [48] models are the most popular, particularly the SWAT model. SWAT includes the statistical model of USLE [40] and derived RUSLE [20]. However, SWAT and APEX are only capable of simulating, mechanistically, a limited number of various best management practices (BMPs) scenarios individually [50].

Recently, due to the climate change which causes the climate variation in some part of the world, particularly rainfall patterns, this will increase and enhance soil erosion, especially in areas where land use changes occur. Rainfall erosivity is the potential ability of rain to cause erosion [51] and it is a major driving force of soil erosion and nutrient losses worldwide, which may leave farmers vulnerable to crop failures. The rainfall erosivity, derived from 30-min or daily rainfall event is characterized by a large variability in space and time [52,53]. This may lead to an inaccurate estimates of soil erosion. However, daily rainfall amount is the simplest erosivity factor and it may poorly explains amount of soil loss [54] because erosivity is also a function of raindrop's diameter, mass, and velocity. In most countries, soil loss measurement is not available. Therefore, an erosivity index cannot be determined empirically [51]. Estimating the impact of future climate change on soil erosion susceptibility can be done by calculating the future predicted R-Factor value. Spatial correlation between climate change, soil erosion and land cover change using global models, such as RUSLE, can effectively assist in the spatial management process [55].

Soil erosion changes in the future can be done by developing modeling scenarios of the two most dynamic factors in soil erosion, i.e., rainfall erosivity and land cover change [53]. Currently, it is believed that large-scale estimation of soil loss rates under climate change conditions is possible [42]. According to Panagos et al. [53], the prediction of soil erosion changes in the future are mainly dependent on modeling future rainfall erosivity, land use changes and impacts of policies on soil loss. Recently, the development of Rainfall Erosivity Database at European Scale (REDES) and statistical methods for spatially interpolating rainfall erosivity data can become valuable insights for predicting future rainfall erosivity based on climate scenarios [52]. Using a comprehensive statistical modeling method (Gaussian Process Regression) will help to predict rainfall erosivity according to climate change scenarios by selecting the most appropriate covariates (monthly precipitation, temperature datasets and bioclimatic layers) [53]. Extreme rainfall will be more intense, and natural disasters will be related to more frequent rainfall; as a result, soil erosion rates are expected to increase in response to climate change [5,42]. Moreover, climatologists have discovered that the earth is warming, and as the global temperature rises, the water cycle becomes more vigorous. Therefore, it is clear that climate change will affect soil erosion and its consequences [56].

In Cambodia, forest cover has declined dramatically in the last few decades, while the research on soil erosion loss caused by LULC change is limited, particularly in the Stung Sangkae catchment, where there is little research on soil erosion reported to date. Most soil erosion and sediment studies were conducted at a large river basin scale such as Mekong River Basin (MRB) or Lower Mekong Basin (LMB) [3,37,57]. Land use in Cambodia began to change due to investments from insiders (Cambodian investors) and outsiders (foreign investors), mainly industrial crops such as palm oil (*Elaeis guineensis*),

rubber (*Hevea brasiliensis*), cassava (*Manihot esculenta*), and kapok (*Ceiba pentandra*) [58]. According to FAO [59], agricultural land expanded from 4580 to 5455 thousand ha (26 to 31% of total land area) from 1997 to 2007. The rice (*Oryza sativa*) production area increased from 2.72 million ha in 2009 to 3.05 million ha in 2013. Similar to the upward trend of the rice production area, the production areas of other four main crops, namely: maize (*Zea mays*), cassava, mung bean (*Vigna radiata*) and soy bean (*Glycine max*), increased from 206,058 to 239,748 ha, 160,326 to 421,375 ha, 49,599 to 54,312 ha, and 96,388 to 80,680 ha from 2009 to 2013, respectively [60], as cited in [61]. According to the Ministry of Environment (MoE) [62], the forest cover of Cambodia declined from 73.04% in 1965 to 48.14% in 2016, compared to the overall country area. This was primarily caused by civil war, population increase, the need of land for agricultural production, and other vital factors. Based on the forest cover assessment, the country's forest cover in 2016 was about 8,742,401 ha (48.14%), and the average annual loss rate from 2014 to 2016 was about 121,328 ha (0.67%), compared to the entire country's area [62].

A recent study by Lohani et al. [63] reported that the primary forest loss in Cambodia from 1993 to 2017 was 17,150 km$^2$, while in Tonle Sap the total area of forest loss was low at 1944 km$^2$; however, when analyzed as a percentage of total forest area of all study regions, this was the highest. The portion of forest land cover in the Tonle Sap was lower as a whole (26 versus 53%) than for the Cambodia and 3S river (Srepok, Sesan, and Sekong) region. The rate of forest loss across the Tonle Sap region was relatively high and constant at 1.2% [63]. Forest loss in the Tonle Sap region seemed to occur over the 25 years and mostly happened in small patches. After 2010, deforested areas have mingled into greater patches. In the western part of Cambodia, there are many large patches of forest loss centered on highland regions, such as the Cardamom Mountains. In contrast, in the north-western part of Cambodia along the border with Thailand, a concentrated area of forest loss occurred in the early 2000s. It decreased only after nearly all remaining primary forests were lost [63].

Based on the finding of Nalin et al. [64], in Tonle Sap Basin, from 1990 to 2009, forest cover decreased by 43% from 20,170 km$^2$ to 11,436 km$^2$, while agricultural land increased by 34% from 14,076 km$^2$ to 18,858 km$^2$. A recent finding by Kong et al. [65] mentioned that the total forest coverage (dense and degraded forests) remained almost unchanged, accounting for about 90% of the area between 1976 and 1997. However, about 13% of the dense forest area was transformed to degraded forestland. Forest cover was reduced dramatically during the following 20 years from 1997 to 2016, and only 25% remained in 2016, especially along the main roads. Sixty-five (65%) percent of the forest cover loss primarily occurred between 2006 and 2016 [65]. Based on the statement of Land Degradation Neutrality Targets (LDNT) in Cambodia, the drivers of land degradation in Cambodia have mainly been attributed to deforestation, expanding agricultural lands, climate change, pest and diseases, unsustainable land management, and infrastructure development. In recent decades, deforestation has resulted in a significant loss of forest cover from 10.83 million ha (59.64%) in 2006 to 8.52 million ha (46.90%) in 2014 to 10.45 million ha (57.55%) in 2010, to 8.52 million ha (46.90%) in 2014 and to 8.22 million ha (45.26%) by 2016. Over that period, croplands (paddy rice fields, field crops, horticulture, rubber and oil palm) increased by about 2.69 million ha. Agricultural land is expanding from lowland to upland, adding more pressure on forestland. Land Productivity Dynamics (LPD) indicate that in 2010, Cambodia had about 53,000 ha of land, showing an early sign of decline in productivity, as land use changed from forest to cropland. The soil organic carbon density indicates that for a period of 10 years (2000–2010), Cambodia lost about 1.98 million tons of carbon in the top 0–30 cm depth due to land use changes from forest to non-forest [66]. Cambodia aims to achieve an economic growth rate of 7% per annum with its aspiration to reach an upper-middle income country by 2030 and is committed to attaining zero-hunger by 2025. Agriculture continues to be a driver of economic growth and poverty reduction for Cambodia. Achieving sustainable agricultural development at 5% per annum is instrumental in addressing the Royal Government of Cambodia's (RGC) objectives for food security, poverty reduction, and increased climate resilience [66]. Food production relies mainly on land and water.

Land degradation and water scarcity are real challenges for food security. As one of the UNCCD (United Nations Convention to Combat Desertification) signatory States, the RGC has approved the National Action Plan (NAP) for 2018 to 2027, a fundamental document for national strategies for combating land degradation in the country. The RGC is committed to achieving 17 Sustainable Development Goals (SDGs), including SDG15, which aims to protect, restore and promote sustainable use of terrestrial ecosystems, sustainably manage forests, combat desertification, and halt and reverse land degradation and halt biodiversity loss. Target 15.3 clearly aims to combat desertification, restore degraded land and soil, including land affected by desertification, drought and floods, and strive to achieve a land degradation-neutral world by 2030 [66].

Thus, the overall goal of this study was to evaluate the land use and land cover changes and its impact on soil erosion in Stung Sangkae catchment in the years 2002 and 2015. The specific objectives of the study were to: (1) estimate the magnitude of annual soil erosion and its spatial distribution in the catchment; and (2) evaluate how land use and land cover types contributed to soil erosion in the catchment. The results of this study are expected to provide useful information that can promote soil erosion management practices in Stung Sangkae Catchment, Battambang Province, as well as Tonle Sap Great Lake, which represents one of the world's most productive ecosystems and biodiversity. The Tonle Sap River-Great Lake system underpins the world's biggest freshwater fishery and directly or indirectly affords a livelihood for most of Cambodia's population [67].

## 2. Materials and Methods

### 2.1. Study Site Description

The Stung Sangkae catchment (605,170 ha), which is the third-largest tributary of the Tonle Sap Basin river system, is located at the upper north-western part of Cambodia between 12°13′–13°24′ N and 102°35′–103°42′ E (Figure 1). The topography is level within the floodplain region and rough with slopes at the upland portion of the catchment having elevations extending from 4 m at the most reduced point to 1413 m a.s.l at the most noteworthy point. The main river that flows through the catchment, Sangkae River, lies between the tributaries of the Tonle Sap Great Lake in the upper western part of the catchments. Agriculture is the main local economic activity and the main source of livelihood. Meteorological data collected from six weather stations in 2007–2018 showed that the average annual precipitation in the study area varied from 1308 mm at Moung Ruessei station to 1577 mm at Samlout station with little change during the year (Figure 2). The major soil types in the region are categorized into 4: (1) Gleysols are wetland soils, which in the natural state are continuously water-saturated within 50 cm of the surface, for extended periods; (2) Luvisols are a type of soil in which highly active clay migrates from the top part of the profile, usually gray, and is deposited in the B layer, usually brown; (3) Nitisols are mainly deep, well-drained soils with a stable structure and high nutrient content; and (4) Acrisols are clay-wealthy soils which can be fairly vulnerable to erosion.

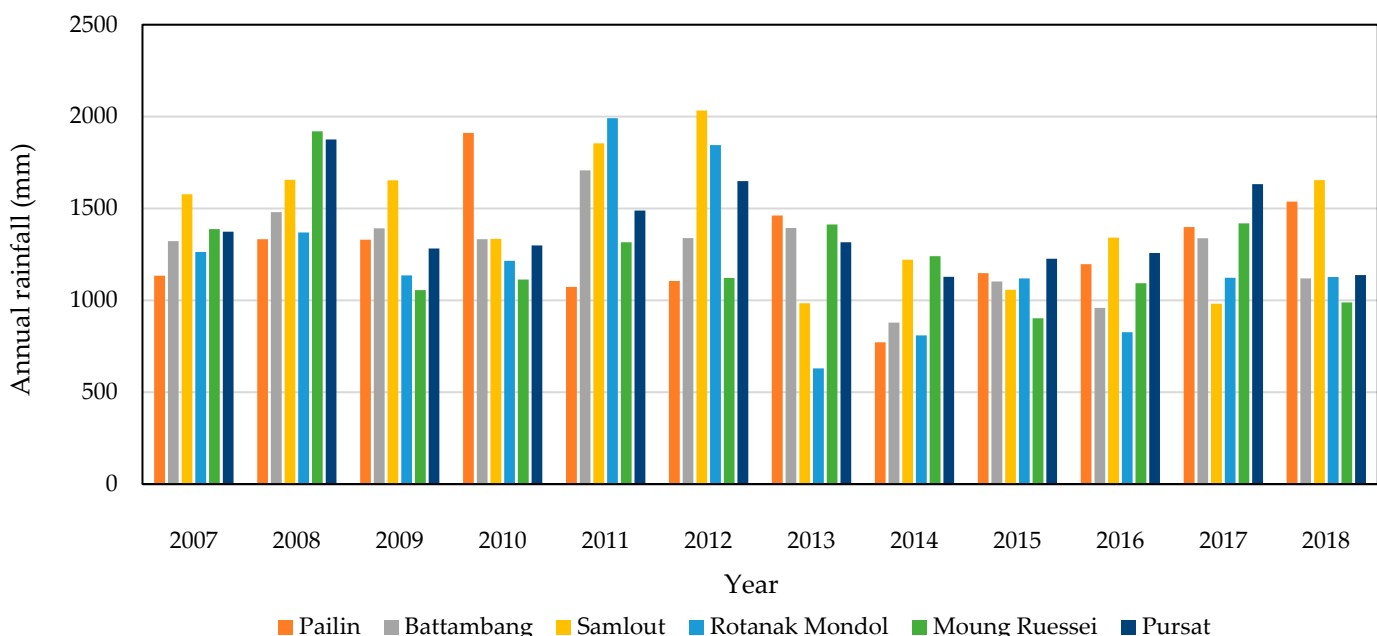

**Figure 1.** Location map of the research catchment and meteorological stations within the research area.

**Figure 2.** The distribution of annual rainfall recorded by weather stations inside and around the study catchment during 2007–2018.

The catchment is characterized by distinctive topographical conditions, from flat plains to rugged areas. After dividing the digital elevation model (DEM) into six FAO slope grades [68]; 16.6% of the total area has very gently sloping (0–2°), whereas 35.2% and 28.3% of the entire areas are characterized as gently sloping (2–5°) and sloping (5–10°). The remaining land slopes are divided into strongly sloping, moderately steep and steep, which covered the areas of 9.8% (10–15°), 9.1% (15–30°) and 1.0% (>30°), respectively (Table 1).

**Table 1.** FAO slope classification in the study catchment and related susceptibility to soil erosion.

| No | Slope Classes (Degree) | Characteristics | Susceptibility | Area (ha) | Area (%) |
|----|----|----|----|----|----|
| 1 | 0–2 | Flat to very gently sloping | Very low | 100,579 | 16.6% |
| 2 | 2–5 | Gently sloping | Low | 212,830 | 35.2% |
| 3 | 5–10 | Sloping | Medium | 171,084 | 28.3% |
| 4 | 10–15 | Strongly sloping | High | 59,375 | 9.8% |
| 5 | 15–30 | Moderately steep | Very high | 55,173 | 9.1% |
| 6 | >30 | Steep | Extremely high | 6129 | 1.0% |

The land use developed by the Japan International Cooperation Agency (JICA) in 2002 [69] and land cover (Land Cover Maps of LMB) developed by Mekong River Commission (MRC) in 2015 [60] were used in the study (see Table 2 and Figure 3). The land cover maps of the Lower Mekong Basin, which covers the Lower Mekong countries such as Cambodia, Lao PDR, Thailand and Vietnam was developed by MRC following the FAO Land Cover Classification System based on the target surveyed points and the satellite image classifications. In Cambodia, the number of target surveyed points were 2595 points [61]. However, due to site conditions, not all points could be inspected; only 9357 points were collected from field data, which accounted for 89% of the target (10,575 points). The samples covered all 19 land cover types. This approach would have resulted in 12,825 samples for the entire LMB. However, the targeted sample size was reduced to 10,575 samples; as a result, only 9357 samples were collected on-site.

**Table 2.** Land use and land cover of the Stung Sangkae catchment in 2002 and 2015.

| LULC Classes | JICA 2002 Area (ha) | JICA 2002 Area (%) | MRC 2015 Area (ha) | MRC 2015 Area (%) | Net Change Area (ha) | Net Change Area (%) |
|----|----|----|----|----|----|----|
| Agricultural land | 25,627.2 | 4.24 | 152,742.3 | 25.24 | 127,115.0 | 21.00 |
| Barren land | 149.2 | 0.02 | 274.0 | 0.04 | 124.8 | 0.02 |
| Built-up area | 1702.8 | 0.28 | 20,870.1 | 3.45 | 19,167.3 | 3.17 |
| Deciduous forest | 74,524.7 | 12.31 | 24,144.9 | 3.99 | −50,379.8 | −8.32 |
| Evergreen forest | 110,474.4 | 18.26 | 90,338.0 | 14.93 | −20,136.4 | −3.33 |
| Grassland | 79,496.0 | 13.14 | 29,394.2 | 4.86 | −50,101.8 | −8.28 |
| Marsh and swamp | 280.3 | 0.05 | 35.8 | 0.01 | −244.6 | −0.04 |
| Mixed forest | 75,361.5 | 12.45 | 64,710.9 | 10.69 | −10,650.6 | −1.76 |
| Paddy field | 92,784.8 | 15.33 | 144,931.5 | 23.95 | 52,146.7 | 8.62 |
| Shrubland | 141,689.0 | 23.41 | 74,019.0 | 12.23 | −67,670.0 | −11.18 |
| Water bodies | 3080.1 | 0.51 | 3709.4 | 0.61 | 629.3 | 0.10 |
| Total | 605,170.0 | 100.00 | 605,170.0 | 100.0 | | |

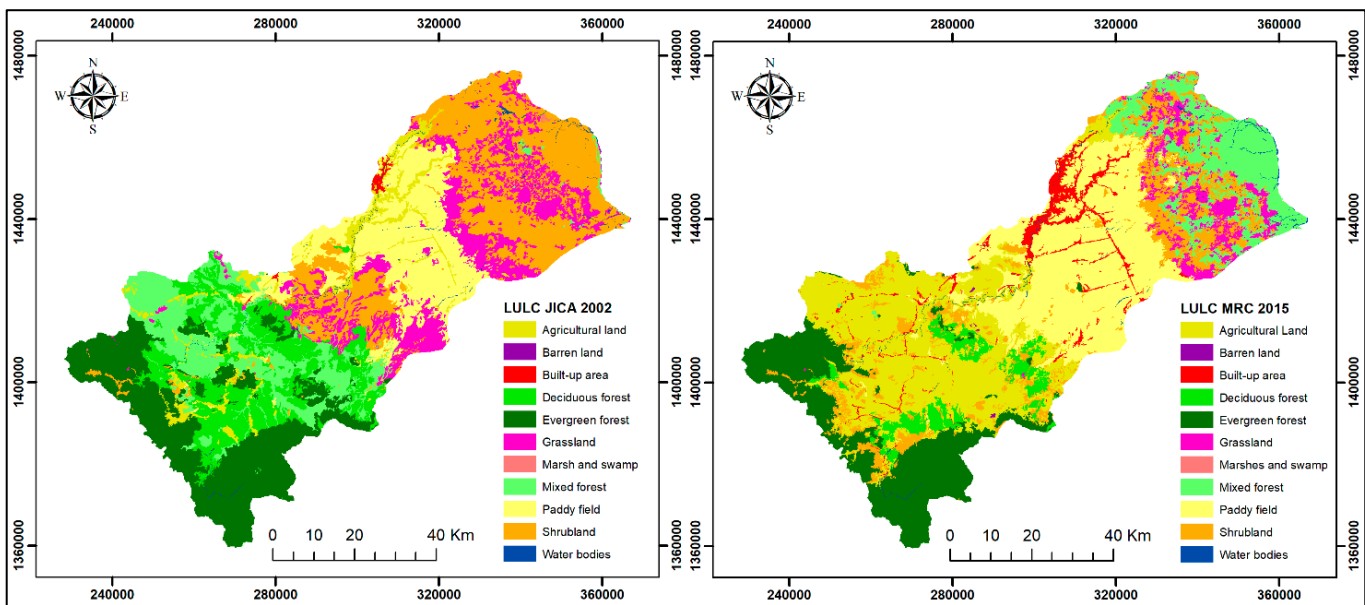

**Figure 3.** Land use and land cover (LULC) developed by JICA 2002 and MRC 2015 of the Stung Sangkae catchment.

From the land use and land cover assessment in Table 2, cultivated lands (agricultural land and paddy rice fields) occupied almost 50% of the total land area in the region in 2015, which increased from 20% in 2002, while forest cover (evergreen, deciduous and mixed forest) occupied 43% in 2002 and declined to 30% in 2015. Among the land use and land covers, areas under agricultural land increased from 4.24% to 25.24%, which is the highest compared to others, followed by paddy rice fields that increased from 15.33% to 23.95% between the years 2002 and 2015. The built-up areas also increased from 0.28% in 2002 to 3.45% in 2015, while water bodies also increased slightly from 0.51% to 0.61% between 2002 and 2015. On the contrary, evergreen forest, deciduous forest, mixed forest, grassland, shrubland and marsh and swamp areas decreased from 18.26%, 12.31%, 12.45%, and 23.41% in 2002 to 14.93%, 3.99%, 10.69%, 4.86% and 12.23% in 2015, respectively.

### 2.2. Determination of RUSLE Factor Values

The applied methodology (as shown in Figure 4) to estimate soil erosion rate in the study catchment was employed with the GIS-based Revised Universal Soil Loss Equation (RUSLE) model [20]. The obtained geospatial input parameters for the RUSLE model (Table 3) were used to produce thematic maps for the estimation of potential soil erosion risk. RUSLE is an extension of the Universal Soil Loss Equation (USLE) model by adjusting the input factors for the local conditions [20,70]. The application of the RUSLE model is simple and applicable in limited data conditions. Because of its suitable capacity and relatively simple computational inputs, RUSLE has been widely used around the world [22,24–28] including in Ethiopia [71]; Kenya [15]; Zimbabwe [9]; China [29,32]; Japan [6]; India [30,31]; Nepal [33,34]; Sri Lanka [72]; South-East Asian countries (Philippines [73–75]; Thailand [76,77]; and Mekong River Basin [3,35,37]. Furthermore, the RUSLE also provides international applicability and comparability for the results and methods, because the model can be adjusted and applied in many parts of the world. The RUSLE model and its predecessor USLE [40] estimate the rate of mean annual soil loss by considering multiple factors expressed in Equation (1):

$$A = R \times K \times LS \times C \times P \tag{1}$$

where: $A$ is the mean annual soil loss (t/ha/y); $R$ is the rainfall erosivity factor (MJ/mm/ha/hr/y); $K$ is the soil erodibility factor (t/hr/MJ/mm); $LS$ is the topographic factor

(dimensionless); *C* is the cropping management factor (dimensionless); and *P* is the support practice factor (dimensionless).

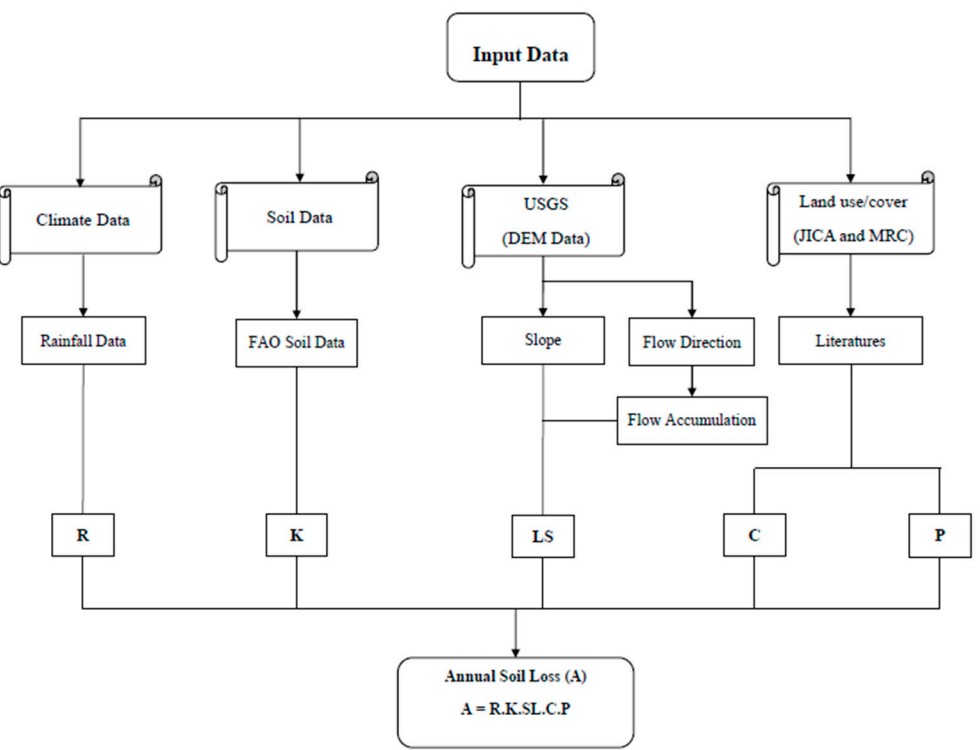

**Figure 4.** Flowchart of applied methodology for modelling soil erosion in the catchment.

**Table 3.** Data used and data source for soil erosion modelling in the catchment.

| No. | Factors | Resolution | Data Source | Format |
|-----|---------|------------|-------------|--------|
| 1 | R Factor | - | Daily rainfall data (2007–2018) from Ministry of Water Resources and Meteorology in Cambodia (MORWAM). | Raster |
| 2 | K Factor | 1 km | FAO/UNESCO Soil Map of the World database through the Harmonized World Soil Database (HWSD) website [78] | Raster |
| 3 | LS Factor | 30 m | Digital Elevation Model (DEM) from the United States Geological Survey (USGS) website [79] | Raster |
| 4 | C Factor | 30 m | Obtained by assigning weighted C factor values to the LULC based on the literatures [3,9,20,22,73,80–82] | Raster |
| 5 | P Factor | 30 m | Obtained by assigning weighted P factor values to the LULC based on the literature as suggested by Yang et al. [81]. | Raster |

### 2.2.1. Rainfall Erosivity (R) Factor

The R-factor accounts for the erosive force of a specific rainfall [40]. The erosive power of a particular precipitation is determined by the amount, intensity and distribution of precipitation; where intensity is the most important property determining the amount of erosion [83]. Therefore, in the original USLE and its revised version (RUSLE), the R-factor was represented in the rainfall intensity data. The annual R-factor is a function of the average annual $EI_{30}$ that is calculated from detailed and long-term records of storm kinetic energy (E) and the 30-min maximum intensity ($I_{30}$) of the storm [20,80]. In general, rainfall intensity data is rarely available in Cambodia, especially in the research areas. For this reason, daily rainfall data collected from six weather stations (Figure 2) in 2007–2018 were obtained from the Ministry of Water Resources and Meteorology (MOWRAM) of Cambodia. Then, the average annual rainfall of the stations (2007–2018), required for the calculation of the R-factor was drawn from the daily data set. The calculated R-factor was interpolated

using the inverse distance weighting (IDW) method and converted into a 30 m cell size grid (Figure 5a).

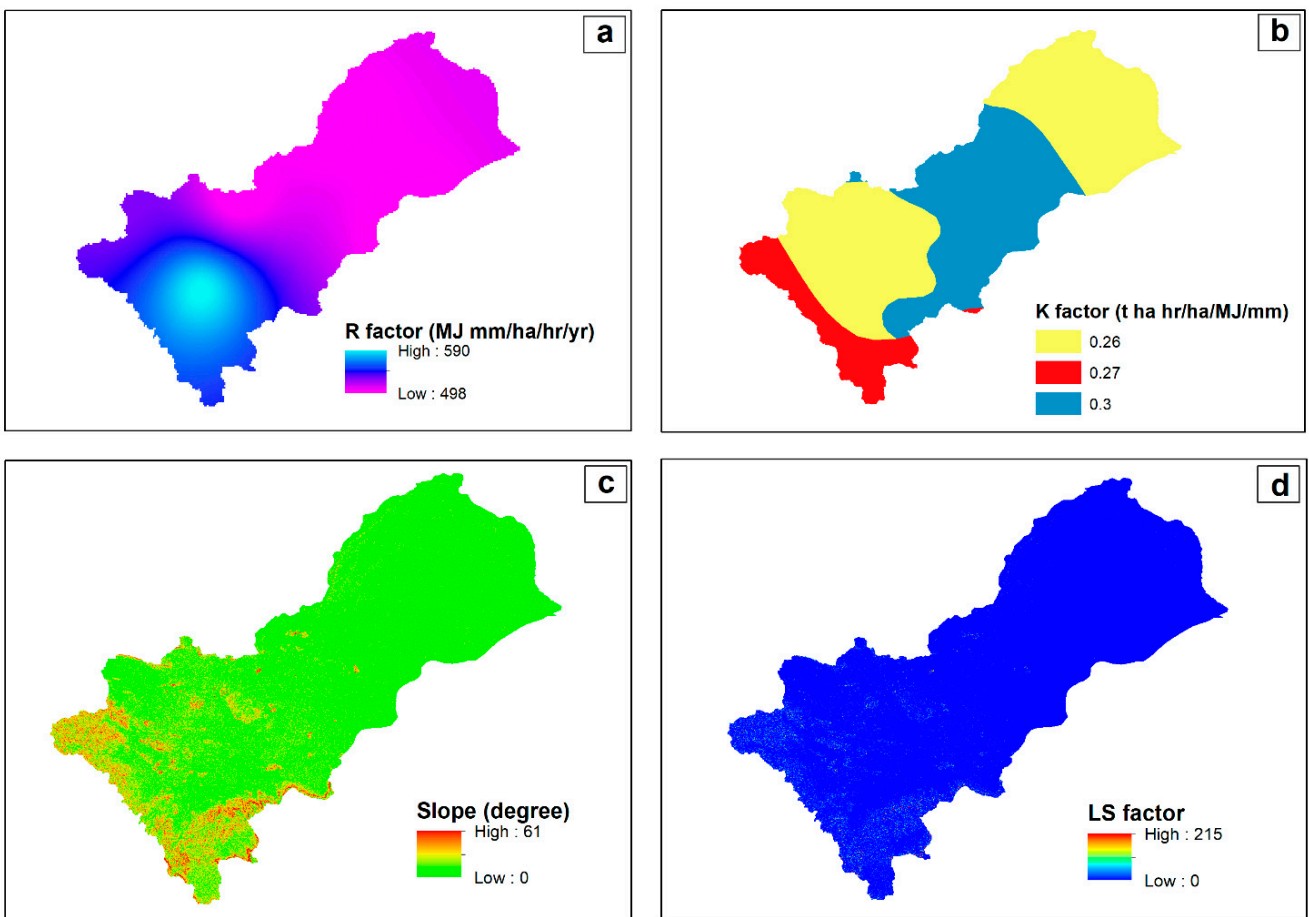

**Figure 5.** Maps of R-factor (**a**), K-factor (**b**), slope (**c**) and LS-factor (**d**) of the Sangkae catchment.

In this study, the equation (Equation (2)) was chosen to calculate the R-factor from Reference [84]. The equation has been adopted by many users in Southeast Asian countries and has been extensively used in Thailand [77,85], Philippines [73–75] and also Sri Lanka [72], Nepal [34] and African countries of Rwanda [86], and Zimbabwe [9].

$$R = 38.5 + 0.35\ P \tag{2}$$

where R = rainfall erosivity (MJ/mm/ha/hr/y) and P = mean annual rainfall amount (mm).

2.2.2. Soil Erodibility (K) Factor

The K-factor corresponds to the influence of the soil's physical and chemical properties on erosion during storm events in upland areas [20,40]. Some of the soil properties that affect soil erodibility include soil texture, drainage condition, soil depth, structural integrity and organic content [30]. Among the different methods for computing the K-factor, the soil nomograph method, which uses the relative ratios of soil texture, permeability, soil structure and organic matter content [40], is the most commonly used method. In this study, soil data were acquired from FAO/UNESCO Soil Map of the World database through the Harmonized World Soil Database (HWSD) because the observed data of the local soil properties in Cambodia is limited and difficult to access. The HWSD is a 30 arc-second raster database (approximately 1 km of spatial resolution) with over 15,000 different soil mapping units that combine existing regional and national updates of soil information around the world. The soil information extracted from the database for assessing soil

erodibility includes sand, silt, clay, and organic carbon. The mentioned soil parameters were used to compute the K-factor based on the following Equations.

$$K_{USLE} = f_{csand} \times f_{cl-si} \times f_{orgC} \times f_{hisand} \tag{3}$$

$$f_{csand} = \left(0.2 + 0.3 \times \exp\left[-0.256 \times m_s \times \left(1 - \frac{m_{silt}}{100}\right)\right]\right) \tag{4}$$

$$f_{cl-si} = \left(\frac{m_{silt}}{m_c + m_{silt}}\right)^{0.3} \tag{5}$$

$$f_{orgC} = \left(1 - \frac{0.25 \times OrgC}{OrgC + \exp[3.72 - (2.95) \times OrgC]}\right) \tag{6}$$

$$f_{hisand} = \left(1 - \frac{0.7 \times \left(1 - \frac{m_s}{100}\right)}{\left(1 - \frac{m_s}{100}\right) + \exp\left[-5.51 + 22.9 \times \left(1 - \frac{m_s}{100}\right)\right]}\right) \tag{7}$$

where $K$ is the soil erodibility factor, $f_{csand}$ is a function of the high coarse sand content of the soil, $f_{cl-si}$ is a function of the clay and silt of the soil, $f_{orgC}$ is a function of the organic carbon content of the soil, $f_{hisand}$ is the function of high sand content in the soil, $m_s$ is the % sand content (0.05–2.00 mm diameter particles), $m_{silt}$ is % silt content (0.002–0.05 mm diameter particles), $m_c$ is the % clay content (<0.002 diameter particles), and $orgC$ is % organic carbon content of the layer (%). The values of the K-factor are between 0 to 1, where values tending towards 1 indicate an increase in susceptibility to erosion by water [86]. The same value of K-factor was used for both LULC of JICA 2002 and MRC 2015 as there were no separate data for the different periods.

### 2.2.3. Topographic (LS) Factor

The topographic factor is one of the most important parameters of the RUSLE model for determining soil erosion since the gravity force plays an important role in surface runoff [87,88]. This factor combines the slope length (L), which measures the distance from the source to the top of the intercalation, and the slope steepness (S). The slope length measurement is incomplete, in which the catchment is characterized as heterogeneous, and considers the topographic scale and aspects related to LULC [87,89]. The LS-factor combines both the length and steepness of the land slope, so it noticeably affects the soil loss rate. This factor was calculated from the DEM of Cambodia at a 30-m spatial resolution obtained from the United States Geological Survey (USGS) Earth Explorer [79]. The LS factor maps were created using ArcGIS 10.3 and the ArcHydro extension tools to undertake DEM sink filling prior to creating the flow direction and flow accumulation. Then, the'surface slope angle was calculated from the DEM, and the LS factor was computed using the following equation as recommended by [89]. This equation has been adopted by several researchers, [15,90–92].

$$LS = \left(Flow\ accumulation \times \frac{cell\ size}{22.13}\right)^m \times \left(0.065 + 0.045s + 0.0065s^2\right)$$

where: *cell size* is the resolution of the DEM pixels (30 m resolution pixel), $s$ is the slope gradient in %; $m$ = dimensionless exponent based on the steepness of the land. The values of m are assigned as: 0.5, 0.4, 0.3 and 0.2 for slopes of >5%, 3–5%, 1–3% and <1%, respectively [20,93]. The same LS-factor was used for both study years of JICA 2002 and MRC 2015.

### 2.2.4. Crop Management (C) Factor and Conservation Practice (P) Factor

The cover and management factor (C) is expressed as the soil loss ratio from an area with a certain cover and management, in which the C-factor accounts for the role of vegetative covers against water erosion [40,94]. In the areas without vegetation, soil erosion by water is high. Conversely, due to the high protection of the soil surface by the vegetation

against erosion, the soil erosion from the land with vegetation cover is low. Therefore, this can reduce soil erosion by returning the LULC types into more vegetation surface covers. For this reason, the C-factor is probably the most crucial factor in reducing soil erosion. An easier way to determine the C-factor is to report similar land cover values and refer to previous studies, or to studies conducted in the same area or region [4]. However, it is important to note that the definition of land cover type may differ among countries when using the C-factor in the literature. For Instance, land classified as a forest in one country may have a different vegetation cover or type than forests in another country (e.g., the difference between a pine forest and a tropical rainforest). Therefore, it is important to understand the differences in land cover classifications before applying the C-factor values from the literature [4]. To develop C-factor maps of the study catchment from the corresponding LULC temporal layers, C factors were assigned for each LULC type based on the literature (Table 4).

**Table 4.** Adopted values of C and P factor for the catchment land use and land cover (LULC) classes.

| LULC Classes | 1C Factor | 2P Factor | References |
|---|---|---|---|
| Agricultural land | 0.5 | 0.5 | 1, 2 [81], 1 [81], 1 [3], 1 [37] |
| Barren land | 0.35 | 1.0 | 1, 2 [81], 2 [20], 2 [80], 2 [20], 2 [22], 2 [9] |
| Built-up area | 0.1 | 1.0 | 1, 2 [81], 2 [20], 2 [80], 2 [20], 2 [22], 2 [9] |
| Deciduous forest | 0.01 | 1.0 | 1, 2 [81], 2 [20], 2 [80], 2 [20], 2 [22], 2 [9] |
| Evergreen forest | 0.001 | 1.0 | 1, 2 [81], 2 [20], 2 [80], 2 [20], 2 [22], 2 [9] |
| Grassland | 0.08 | 1.0 | 1, 2 [81], 2 [20], 2 [80], 2 [20], 2 [22], 2 [9] |
| Marsh and swamp | 0.05 | 1.0 | 1, 2 [81], 2 [20], 2 [80], 2 [20], 2 [22], 2 [9] |
| Mixed forest | 0.1 | 0.8 | 1, 2 [81] |
| Paddy field | 0.1 | 0.5 | 1, 2 [81], 1 [82], 1 [80], |
| Shrubland | 0.014 | 1.0 | 1, 2 [81], 2 [20], 2 [80], 2 [20], 2 [22], 2 [9] |
| Water bodies | 0.01 | 1.0 | 1, 2 [81], 2 [20], 2 [80], 2 [20], 2 [22], 2 [9] |

The P-factor represents the role of conservation practices in reducing erosion [40]. The value of P-factor is between 0 and 1. In general, 1 is assigned to areas without protection measures [22,73], and a minimum value close to 0 is given for areas with suitable protection measures. Therefore, the lower the P value is, the more effective the protection against erosion is [30]. A review of the RUSLE model by Benavidez et al. [4] emphasized that the P-factor could also be estimated using sub-factors. Even so, the difficulty of accurately mapping supporting practice factors or not observing support practices has led many studies to ignore it by setting the value of its P-factor to 1, as seen in other studies [9,22,73]. However, in the studied catchment, the P factor was determined based on the land cover type from the C-factor (Table 4) as suggested by Yang et al. [81].

## 3. Results

### 3.1. RUSLE Factors

The various RUSLE factors identified in this study are shown in Table 5 and Figure 5. The rainfall erosivity (R-factor) value ranged from 496 to 590 MJ mm/ha/hr/y (mean of 524). The rainfall erosivity factor map for Stung Sangkae catchment depicts moderate variations over the study periods between 2002 and 2015. In parts of the lowland areas, the value of R-factor was below 500 MJ mm/ha/hr/y, while in the upland parts of the catchment, the R-factor was higher reaching up to almost 600 MJ mm/ha/hr/y (Figure 5a).

**Table 5.** The mean annual precipitation (mm) in the study area and the corresponding R-factor.

| Station | Location | | Elevation (m) | Mean Annual Rainfall (2007–2018) | R Factor (MJ/mm/ha/hr/y) |
| --- | --- | --- | --- | --- | --- |
| | Longitude | Latitude | | | |
| Pailin | 102.6115 | 12.85589 | 95 | 1399.8 | 528.4 |
| Battambang | 103.204 | 13.0989 | 94 | 1318.7 | 500.1 |
| Samlout | 102.8594 | 12.61453 | 153 | 1576.9 | 590.4 |
| Rotanak Mondol | 102.9674 | 12.89267 | 258 | 1313.1 | 498.1 |
| Moung Ruessei | 103.4457 | 12.77753 | 29 | 1308.3 | 496.4 |
| Pursat | 103.5400 | 12.3300 | 22 | 1410.7 | 532.3 |

Soils in the catchment upland areas were dominated by Nitosols (clay), which covers 27% of the catchment and Acrisols (clayey loams), which covered 12% of the catchment, while in the lowland and floodplain areas, the soil was dominated by Luvisols (34%) and Gleysols (12%). Thus, the soils varied from clay to clay loams in the catchment based on the soil texture classification (Table 6). The soil erodibility (K-factor) values ranged from 0.26 to 0.3 tons h/MJ/mm (Figure 5b). The slope in the catchment varied from 0–61 degrees, and the LS factor values ranged from 0 to 215 (Figure 5c,d).

**Table 6.** The soil types and the corresponding K-factor in the study catchment.

| Soil Type | Soil Texture | K Factor (t ha h/ha/MJ/mm) | Area (ha) | (%) | References |
| --- | --- | --- | --- | --- | --- |
| Eutric Gleysols (Ge) | Clay | 0.26 | 164,959 | 27% | [81] |
| Gleyic Luvisols (Lg) | Clay Loam | 0.30 | 204,534 | 34% | [81] |
| Dystric Nitosols (Nd) | Clay | 0.26 | 165,639 | 27% | [81] |
| Orthic Acrisols (Ao) | Clay Loam | 0.27 | 70,040 | 12% | [81] |
| | Total | | 605,170 | 100% | |

The values of land cover management factor (C-factor) and the values of conservation practice (P-factor) were based on the literatures. As shown in Figure 6, the spatial distribution of the value of C-factors was in the range of 0.001 to 0.5, while the value of P-factor ranged from 0.5 to 1 (Figure 6).

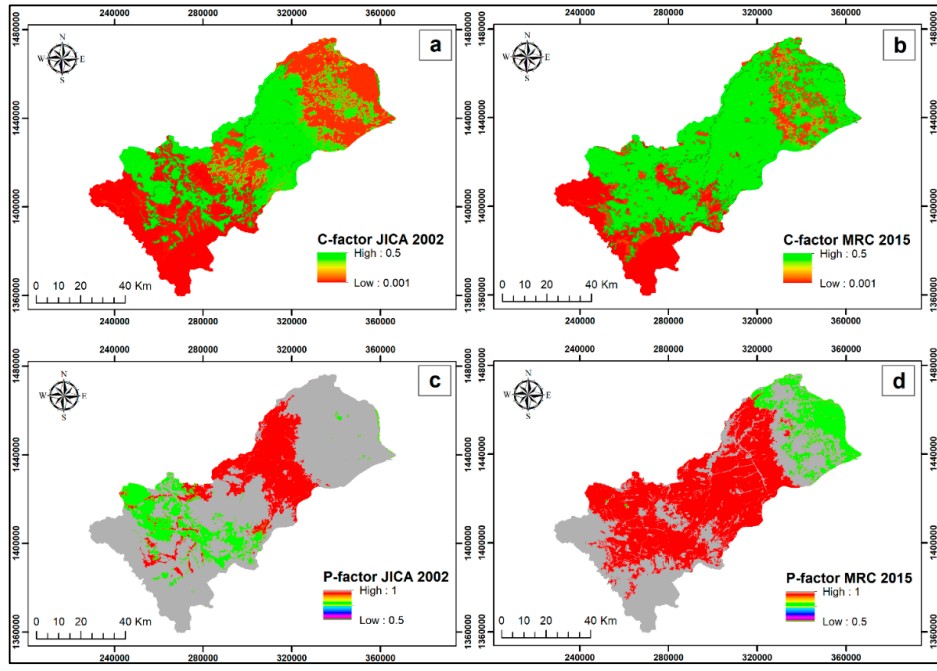

**Figure 6.** Spatial distribution of: (**a**,**b**) cover management (C-factor) and (**c**,**d**) supporting practices (P-factor).

### 3.2. Impact of LULC Changes on Soil Erosion

The RUSLE factors were multiplied in ArcGIS 10.3 spatial analyst tool (zonal statistic) to get the spatiotemporal variations in annual soil erosion rate for the period 2002–2015, and the results are provided in Tables 7 and 8 and Figure 7. The study revealed that the soil loss caused by sheet and rill erosion in Stung Sangkae catchment was in the range of 0–60 t/ha/y in 2002 to 0–63 t/ha/y in 2015 (Table 7). In 2002, the total soil loss was 1,604,234 tons distributed over the catchment, while the amount of total soil loss in 2015 was 3,343,216 tons (Table 8), which happened mainly in the upland of the catchment (Figure 7).

**Table 7.** Distribution of soil erosion loss under different severity classes in Sangkae catchment from 2002 to 2015.

| Severity Classes | Soil Loss (t/ha/y) | JICA 2002 | | | | MRC 2015 | | | Net Change (ha) |
|---|---|---|---|---|---|---|---|---|---|
| | | Area | | Soil Loss (t/ha/y) | | Area | | Soil Loss (t/ha/y) | |
| | | (ha) | (%) | | | (ha) | (%) | | |
| Very low | <2 | 484,089 | 80.0 | 0.2 | | 443,439 | 73.3 | 0.2 | −40,650 |
| Low | 2–5 | 52,646 | 8.7 | 3.2 | | 50,897 | 8.4 | 3.3 | −1749 |
| Moderate | 5–10 | 28,854 | 4.8 | 7.0 | | 33,416 | 5.5 | 7.1 | +4562 |
| Severe | 10–20 | 18,961 | 3.1 | 13.9 | | 25,023 | 4.1 | 14.3 | +6062 |
| Very severe | 20–40 | 11,463 | 1.9 | 27.5 | | 22,940 | 3.8 | 28.5 | −11,477 |
| Extremely Severe | >40 | 9157 | 1.5 | 60.0 | | 29,455 | 4.9 | 62.9 | −20,298 |
| Total Area | | 605,170 | 100.0 | | | 605,170 | 100.0 | | |

**Table 8.** Soil erosion severity classes and gross soil loss in Stung Sangkae catchment from 2002 to 2015.

| Severity Classes | Soil Loss (t/ha/y) | JICA 2002 | | MRC 2015 | | Total Annual Soil Loss | | | |
|---|---|---|---|---|---|---|---|---|---|
| | | Area | | Area | | 2002 | | 2015 | |
| | | (ha) | (%) | (ha) | (%) | (tons) | (%) | (tons) | (%) |
| Very low | <2 | 484,089 | 80.0 | 443,439 | 73.3 | 104,958 | 6.5 | 77,615 | 2.3 |
| Low | 2–5 | 52,646 | 8.7 | 50,897 | 8.4 | 169,155 | 10.5 | 166,630 | 5.0 |
| Moderate | 5–10 | 28,854 | 4.8 | 33,416 | 5.5 | 201,419 | 12.6 | 237,254 | 7.1 |
| Severe | 10–20 | 18,961 | 3.1 | 25,023 | 4.1 | 263,691 | 16.4 | 357,164 | 10.7 |
| Very severe | 20–40 | 11,463 | 1.9 | 22,940 | 3.8 | 315,430 | 19.7 | 653,124 | 19.5 |
| Extremely Severe | >40 | 9157 | 1.5 | 29,455 | 4.9 | 549,581 | 34.3 | 1,851,429 | 55.4 |
| Total Area | | 605,170 | 100.0 | 605,170 | 100.0 | 1,604,234 | 100.0 | 3,343,216 | 100.0 |

The estimated rate of soil erosion was categorized into five severity classes such as very low (0–2 t/ha/y), low (2–5 t/ha/y), moderate (5–10 t/ha/y), severe (10–20 t/ha/y), very severe (20–40 t/ha/y) and extremely severe (>40 t/ha/y), as shown in Tables 7 and 8. The results show that the areas experienced very low erosion rates, which were dominant in the study area, covering 484,089 ha (80.0%) that the average soil loss was 0.2 t/ha/y and 443,439 ha (73.3%) that the average soil loss was 0.2 t/ha/y in the years 2002 and 2015, respectively, while the areas affected by the extreme erosion was 9157 ha (1.5%) and 29,455 ha (4.9%) with the average soil loss of 60 t/ha/y and 63 t/ha/y in the years 2002 and 2015, respectively. In 2002, the areas that experienced moderate erosion (7 t/ha/y), severe erosion (13.9 t/ha/y) and very severe erosion (27.5 t/ha/y) were 28,854 ha, 18,961 ha and 11,463 ha, respectively. In 2015, areas that experienced moderate erosion (7.1 t/ha/y), severe erosion (14.3 t/ha/y) and very severe erosion (28.5 t/ha/y) were 33,416 ha, 25,023 ha and 22,940 ha, respectively.

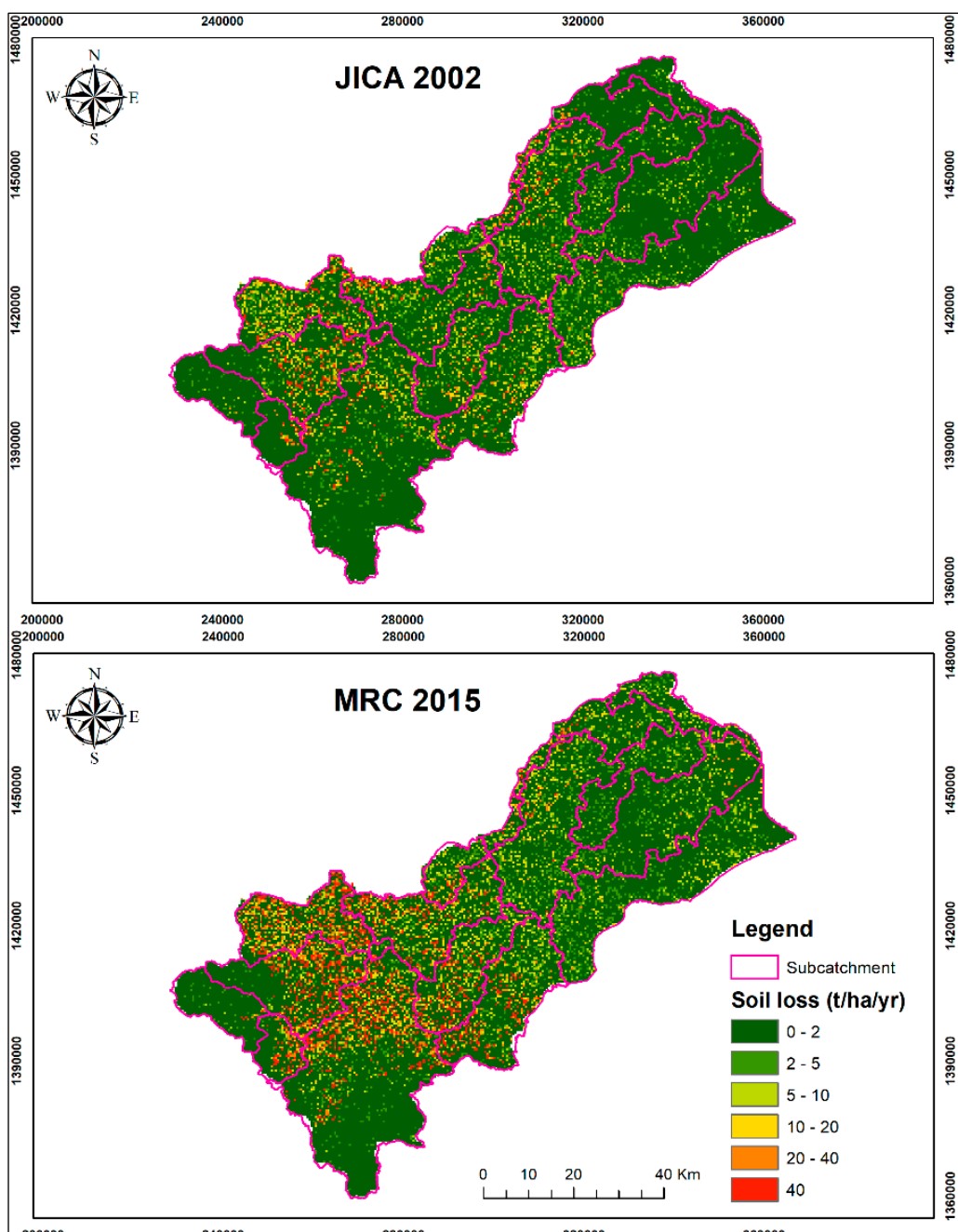

**Figure 7.** Spatial distribution of soil loss in the sub-catchment within the study area.

### 3.3. Effect of Elevation and Slope on Soil Erosion

The elevation of the study catchment was divided into five different zoning areas, and the corresponding soil erosion rates were computed accordingly (Table 9). The soil loss rate at elevations less than 300 m (529,855 ha) was 2.7 t/ha/y in 2002 and 5.2 t/ha/y in 2015, and the change was the highest soil loss amounts to 2.5 t/ha/y. The rate of soil loss for the elevation of 300–600 m (39,347 ha) was 0.8 t/ha/y in 2002 and 0.9 t/ha/y in 2015. In contrast, the rate of soil loss for elevation of 600–900 m (29,454 ha) and 900–1200 m (6065 ha) was 0.4 t/ha/y and 0.9 t/ha/y in 2002 and 0.3 t/ha/y and 0.7 t/ha/y in 2015. The results indicated a decrease in soil loss of about 0.1 t/ha/y and 0.3 t/ha/y that was realized at elevations of 600–900 m (29,454 ha) and those below 1200 m (6065 ha), respectively, while at the elevation of 1200–1500 m (449 ha), the amount of soil erosion was not changed (0.4 t/ha/y).

**Table 9.** Estimation of soil erosion rates and net changes in different elevation areas.

| No | Elevation (Meters) | Area | | Erosion (t/ha/y) | | Net Change (t/ha/y) |
|---|---|---|---|---|---|---|
| | | (ha) | (%) | 2002 | 2015 | |
| 1 | 0–300 | 529,855 | 87.6 | 2.7 | 5.2 | 2.5 |
| 2 | 300–600 | 39,347 | 6.5 | 0.8 | 0.9 | 0.1 |
| 3 | 600–900 | 29,454 | 4.8 | 0.4 | 0.3 | −0.1 |
| 4 | 900–1200 | 6065 | 1.0 | 0.9 | 0.7 | −0.3 |
| 5 | 1200–1500 | 449 | 0.1 | 0.4 | 0.4 | 0.0 |

The amount of soil loss was additionally distributed according to the slope of occurrence (Table 10). The soil erosion rate increased with the increase of slope. The lowest was 0.8 t/ha/y in 2002 and 1.7 t/ha/y in 2015, which occurred in slopes that were less than 2° (100,579 ha). In slopes of 2–5° (212,830 ha), the soil loss rates were 1.7 t/ha/y in 2002 and 3.6 t/ha/y in 2015. In addition, the soil loss rates of slopes of 5–10° (171,084 ha) were 3.6 t/ha/y in 2002 and 9.6 t/ha/y in 2015, while the slope of 10–15° (59,375 ha) was 6.4 t/ha/y in 2002 and 17.7 t/ha/y in 2015. For the slopes of 15–30° (55,173 ha), was 7.2 t/ha/y in 2002 and 16.1 t/ha/y in 2015. Slopes of more than 30° (6129 ha) had soil erosion rates of 16.3 t/ha/y in 2002 and 27.6 t/ha/y in 2015.

**Table 10.** Soil erosion in slope zones and net changes between the years 2002 and 2015 based on FAO slope classification.

| No | Slope Classes (Degree) | Area | | Erosion (t/ha/y) | | Net Change (t/ha/y) |
|---|---|---|---|---|---|---|
| | | (ha) | (%) | 2002 | 2015 | |
| 1 | 0–2 | 100,579 | 16.6% | 0.8 | 1.7 | 0.8 |
| 2 | 2–5 | 212,830 | 35.2% | 1.7 | 3.6 | 1.9 |
| 3 | 5–10 | 171,084 | 28.3% | 3.6 | 9.6 | 6.0 |
| 4 | 10–15 | 59,375 | 9.8% | 6.4 | 17.7 | 11.3 |
| 5 | 15–30 | 55,173 | 9.1% | 7.2 | 16.1 | 8.9 |
| 6 | >30 | 6129 | 1.0% | 16.3 | 27.6 | 11.3 |

*3.4. Contribution of Land Use and Land Cover Changes to Soil Erosion and Its Conversions*

The results indicated that under LULC conditions in 2002, it is estimated that about 1,903,554 tons of soil were lost, while the estimated average soil loss in 2015 was 4,538,331 tons (Table 11). The results also revealed that the amount of soil loss increased almost twice during the investigated periods. For the agricultural land areas in Stung Sangkae catchment, averagely 463,962 tons (24.6%) of soil loss was estimated for 2002, while an increase of up to 3,757,018 tons (81.5%) of soil loss was estimated for the same land use type for 2015. Likewise, the estimated soil loss of land use types of "barren land" and "built-up area" increased slightly from 1240 tons (0.1%) in 2002 to 51,823 tons (1.2%) in 2015 and from 14,748 tons (1.2%) in 2002 to 147,967 tons (3.2%) in 2015. In contrast, estimates of soil loss for land use types such as deciduous forests, evergreen forests, grasslands, mixed forests, and paddy field and so on were decreased. Particularly there was a significant decline of soil loss for the land use of grassland where the estimated soil loss was 241,922 tons (12.5%) in 2002 to 49,179 tons (1.0%) in 2015; and for the mixed forest it decreased from 797,562 tons (41.9%) in 2002 to 129,349 tons (2.8%) in 2015. For the land use types of deciduous and evergreen forest, the estimated soil losses were 83,370 tons (4.4%) and 37,189 tons (1.9%) in 2002 and 28,244 tons (0.6%) and 33,443 tons (0.7%) in 2015, respectively.

**Table 11.** Distribution of soil erosion loss under various types of land use and land cover in Stung Sangkae catchment.

| LULC Classes | JICA 2002 | | | MRC 2015 | | |
|---|---|---|---|---|---|---|
| | Soil Loss (tons) | Area (ha) | (%) | Soil Loss (tons) | Area (ha) | (%) |
| Agricultural land | 463,962 (24.6%) | 25,627.2 | 4.24 | 3,757,018 (81.5%) | 152,742.3 | 25.24 |
| Barren land | 1240 (0.1%) | 149.2 | 0.02 | 51,823 (1.2%) | 274.0 | 0.04 |
| Built-up area | 14,748 (0.7%) | 1702.8 | 0.28 | 147,967 (3.2%) | 20,870.1 | 3.45 |
| Deciduous forest | 83,370 (4.4%) | 74,524.7 | 12.31 | 28,244 (0.6%) | 24,144.9 | 3.99 |
| Evergreen forest | 37,189 (1.9%) | 11,0474.4 | 18.26 | 33,443 (0.7%) | 90,338.0 | 14.93 |
| Grassland | 241,922 (12.5%) | 79,496.0 | 13.14 | 49,179 (1.0%) | 29,394.2 | 4.86 |
| Marsh and swamp | 305 (0.1) | 280.3 | 0.05 | 73 (0.1) | 35.8 | 0.01 |
| Mixed forest | 797,562 (41.9%) | 75,361.5 | 12.45 | 129,349 (2.8%) | 64,710.9 | 10.69 |
| Paddy field | 185,115 (9.7%) | 92,784.8 | 15.33 | 234,464 (6.3%) | 144,931.5 | 23.95 |
| Shrubland | 78,143 (4.1%) | 141,689.0 | 23.41 | 106,771 (2.6%) | 74,019.0 | 12.23 |
| Water bodies | 0 | 3080.1 | 0.51 | 0 | 3709.4 | 0.61 |
| Total Area | 1,903,554 | 605,170.0 | 100.0 | 4,538,331 | 605,170.0 | 100.0 |

The soil loss distribution of different types of LULC showed that the amount of soil loss from agricultural land increased the most, from 463,962 tons (24.6%) to 3,757,018 tons (81.5%). In comparison, soil erosion in the built-up areas also increased dramatically from 14,748 tons (0.7%) in 2002 to 147,967 tons (3.2%) in 2015 (Table 11). In contrast, other LULC types that had significantly lower soil loss were forest lands, paddy field and grass/shrubland, accounting for 48.2%, 4.1%, and 9.7% of soil loss in 2002 and 6.3%, 16.6%, and 3.7% in 2015, respectively (Figure 8).

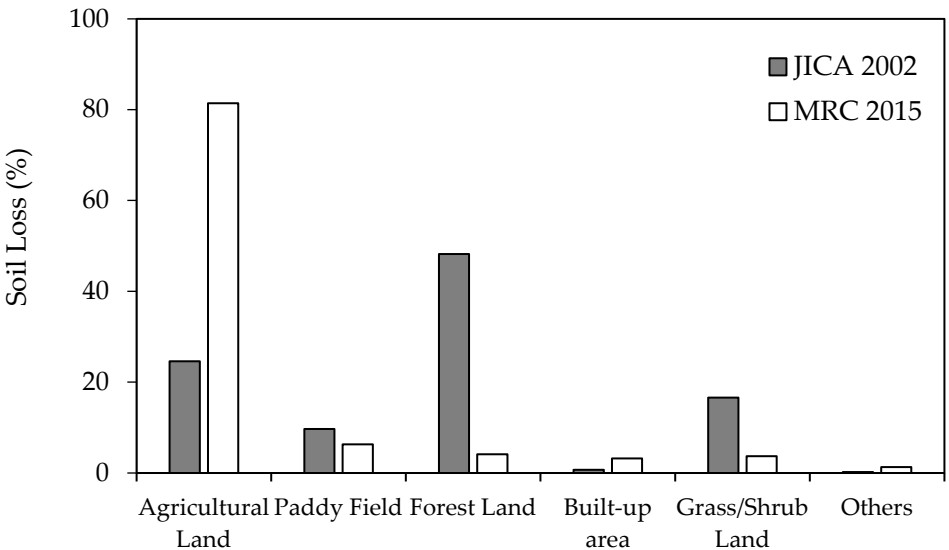

**Figure 8.** Total soil erosion contributed by various land use and land cover types.

The spatial distribution of LULC conversion and its contribution to soil erosion are shown in Figure 9 and Table 12. It should be noted that each LULC category were converted to different land use types during the studied period between 2002 and 2015. However, Table 12 presents only the major LULC conversions from the LULC categories. The primary conversion of LULC changes are agricultural land, shrubland and paddy field, while the highest soil erosion rate happened to agricultural land ranging from 15.6 to 31.3 t/ha, and the soil erosion rate of shrubland is from 0.5 to 4.8 t/ha, while paddy field ranges from 1.9 to 3.0 t/ha. The mean soil erosion rate of the converted LULC varied from 0.3 to 22 t/ha

(Figure 9). All LULC categories were changed to agricultural land which occupied the area of 125,546 ha which mainly converted from mixed forest (57,862.7 ha), deciduous forest (37,911.5 ha), evergreen forest (19,309.7 ha), shrubland (14,775.3 ha), grassland (6348 ha) and paddy field (4114 ha); resulted in total soil loss of 1,597,728.4 ha, 878,981.9 ha, 603,570.1 ha, 117,116.3 ha, 258,200.8 ha and 64,104.8 ha, respectively.

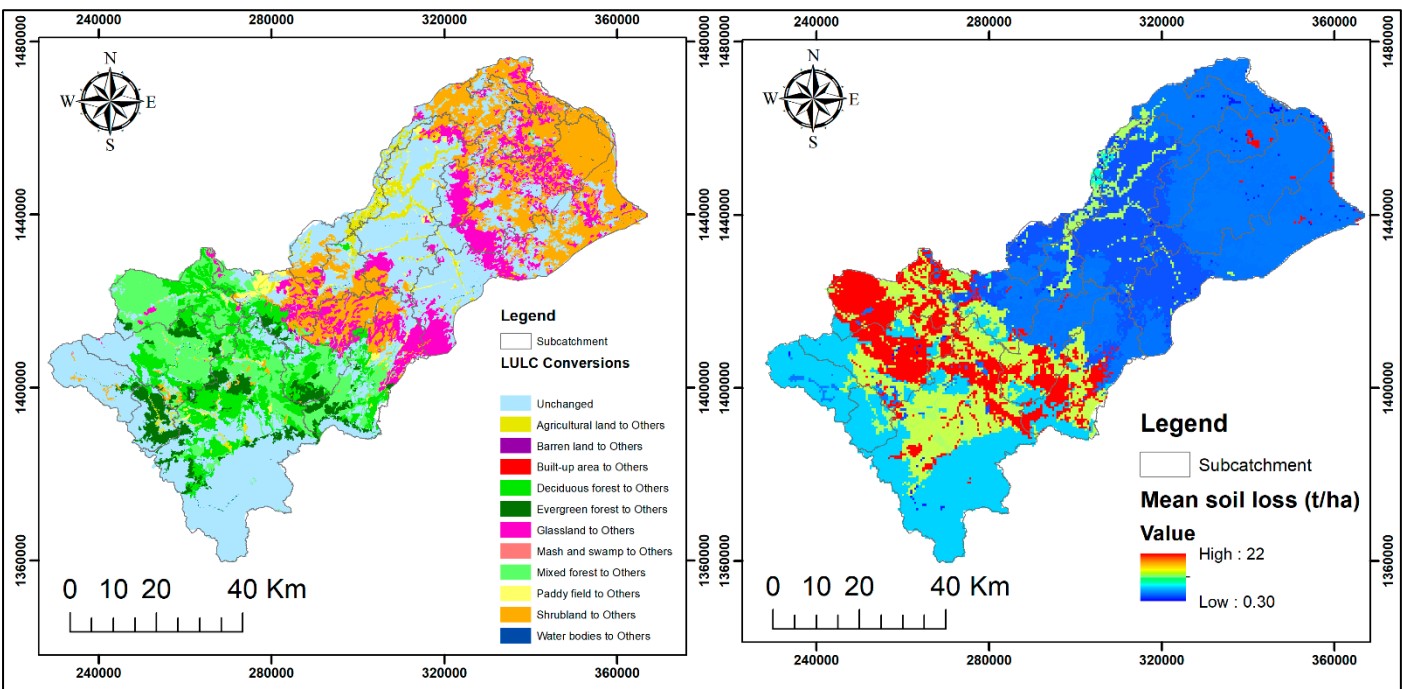

**Figure 9.** Spatial distribution of the conversions of the land use and cover (LULC) between 2002 and 2015.

**Table 12.** Distribution of soil erosion loss under different LULC conversion categories in the in Stung Sangkae catchment.

| No | LULC Categories | Unchanged Area (ha) | Changed Area (ha) | Major LULC Conversions | Changed Area from LULC Categories (ha) | Soil Erosion (t/ha) | Soil Erosion (tons) |
|---|---|---|---|---|---|---|---|
| 1 | Agricultural land | 11,299 | 14,328 | Built-up area | 11,789.30 | 7.2 | 85,319.6 |
| | | | | Paddy field | 1371.36 | 3.0 | 4083.8 |
| 2 | Deciduous forest | 20,712 | 53,813 | Agricultural land | 37,911.56 | 23.2 | 878,981.9 |
| | | | | Shrubland | 9688.02 | 2.3 | 22,501.6 |
| | | | | Evergreen forest | 3583.24 | 0.2 | 611.5 |
| 3 | Evergreen forest | 83,150 | 27,324 | Agricultural land | 19,309.68 | 31.3 | 603,570.1 |
| | | | | Shrubland | 6878.94 | 4.8 | 32,931.1 |
| | | | | Paddy field | 35,168.84 | 1.9 | 67,545.0 |
| 4 | Grassland | 12,396 | 67,100 | Shrubland | 18,624.00 | 0.5 | 9767.2 |
| | | | | Agricultural land | 6348.09 | 18.4 | 117,116.3 |
| 5 | Mixed forest | 1462 | 73,900 | Agricultural land | 57,862.69 | 27.6 | 1,597,728.4 |
| | | | | Shrubland | 9621.66 | 3.8 | 36,994.1 |
| 6 | Paddy field | 82,643 | 10,142 | Built-up area | 4777.65 | 5.6 | 26,819.6 |
| | | | | Agricultural land | 4114.09 | 15.6 | 64,104.8 |
| | | | | Mixed forest | 56,712.51 | 2.1 | 116,896.8 |
| 7 | Shrubland | 27,628 | 114,061 | Paddy field | 23,401.65 | 2.7 | 62,869.5 |
| | | | | Grassland | 16,699.67 | 1.8 | 30,334.7 |
| | | | | Agricultural land | 14,775.34 | 17.5 | 258,200.8 |

## 4. Discussions

This study applied the empirical RUSLE model to predict potential soil erosion in Stung Sangkae catchment, northwestern part of Cambodia, by integrating the RUSLE model into a GIS application based on the national LULC changes in the catchment between 2002 and 2015. As there is no available report or research on soil erosion with the RUSLE model applied in Cambodia, particularly to the catchment, the computation of the potential soil erosion is based on the literature on the assumption of land management and supporting practice; potential erosion is understood as the erosion processes that are only controlled by physical factors [9]. The R-factor is the main driver of soil erosion. There are many equations to estimate the rainfall erosivity factor based on the preferences of the individual researchers and the regions.

A review of the RUSLE model [4] showed that due to the detailed data requirements for the standard (R)USLE computation of rainfall erosivity, alternative equations have been used when studying in areas with less detailed data, depending on the temporal resolution and availability of the precipitation data. In the study, the equation recommended by El-Swaify et al. [84], adopted by many researchers around the world, mostly in Africa (e.g., Ethiopia, Kenya, Zimbabwe) and the Asia (e.g., China, India, Malaysia, Nepal, Thailand, Philippine), particularly the South-East Asian countries and MRB countries were chosen for soil erosion analysis. At the same time, the K-factor was calculated following the equation [95]. According to Yang et al. [96], soil loss is proportional to rainfall erosivity index when all the other factors are held constant; therefore, it is an important factor in the model. The study showed that the spatial distribution of rainfall-runoff erosivity in the catchment was consistent with the amount of precipitation received in various parts of the study catchment. The highest calculated erosivity indices were more in the southwestern regions of the study area, mainly in Phnom Samkos Wildlife Sanctuary, compared with central areas and floodplain areas (Figure 5). In Cambodia, the average annual rainfall is 1400 mm in the central lowland regions and can reach 4000 mm in some coastal areas or in the highlands [97]. As a result, the high rainfall erosivity indices in the region are more likely to occur during the rainy season which runs from mid-May to early October.

The study also determined that the highest erodibility values (Figure 5) were found in the upper regions of the catchment. This indicates that the soils in these areas have stability and low infiltration rates; therefore, they are susceptible to erosion in the event of large flows. The soil erosion rates between 0.2 and 62.9 t/ha/y (Table 7) estimated for the catchment were within similar studies carried out in the MRB. According to Chuenchum et al. [37], in the Lancang MRB, soil erosion loss was mainly classified as moderate erosion in 45% of the study area. Furthermore, in the area around Tenle Sap's soil erosion, it was found that its erosion level was extreme, with more than 80 t/ha/y [37]. Chuenchum et al. [37] reported that the soil erosion of the lower MRB was 198.2 t/km$^2$/y (1.9 t/ha/y), which represents approximately 64% of the total occurrence of soil erosion in the MRB. However, the results of Chuenchum et al. [37] were close to the average values from the previous studies [35], where average soil erosion was found to be between 1400 to 8500 t/km$^2$/y. The differences in these findings may be mainly because of R-factor and LS-factor values, as Chuenchum et al. [37] found that the values of R-factor and LS-factor were 65.6–524.3 MJ.mm/(ha·hr·y) and LS-factor were in the range of 0–336. Meanwhile, Thuy et al. [35] found that the R-factor was 1886–9725 MJ.mm/(ha·hr·y), and LS-factor was from 0.001 to 31.9. Kogo et al. [15] emphasized that due to the variability of topographic features, erodibility, erosivity, and vegetation entrances, the estimated soil erosion rate varies between regions. Based on the Marondedze et al. [9], in the tropical condition, the average soil loss rates of 5/t/ha/y were found in the previous studies [98,99] while it also mentioned that a soil loss limit could be 11t/ha/y accepted as reasonably average annual loss due to soil erosion. However, Hudson [100] believes that for sensitive and fragile lands, the rate of average soil loss tolerance of 2 t/ha/y can be recommended. Additionally, the potential and actual case studies of soil erosion have verified the sensitivity of the C and the P-factors to soil erosion. Natural vegetation covers, such as the forests

(ever-green forest, deciduous forest, and mixed forest) in catchment decreased dramatically around 50,379.8 ha (8.32%), 20,136.4 ha (3.33%), and 10,650.6 ha (1.76%) from 2002 to 2015 (Tables 2 and 11 and Figure 3). Therefore, if forest area is converted into agricultural lands, the rate of soil erosion will increase significantly, especially in the upper reaches [101]. However, it is reported that the RUSLE model lacks the ability to calculate soil losses caused by gully or river channel erosion caused by raindrops [20,40]. Hence, it should be considered that the soil erosion rates found in this study mainly comes from sheet, rill (produced by runoff) and inter-rill (affected by raindrops on the ground) erosion. However, these are the most common processes leading to extensive soil loss in farmland [102].

In terms of the severity classes of soil loss, the results illustrated that 76.6% of the study areas experienced a very low rate of severe soil erosion. Cumulatively, the annual contribution of the low severe soil erosion class is highest due to the expansive extent of their occurrence. These areas cannot be ignored in the agricultural management of soil erosion, because soil loss in these areas will systematically reduce soil quality by removing silt, clay, and organic components that play a vital role in keeping the soil water holding capacity and structural integrity [103].

We also estimated gross soil erosion in the catchment. The results showed significant change in mean soil erosion due to LULC changes during the investigated periods between 2002 and 2015 LULC, in which agricultural land showed a significant increase. In contrast, forest lands (ever-green forest, deciduous forest and mixed forest), grassland and shrubland declined significantly. Soil erosion was considerably higher on cropland (agricultural land and paddy field), built-up area, shrubland and barren land, and low in forested areas (ever-green forest, deciduous forest and mixed forest) and grassland.

The relationship between LULC and estimated soil erosion was analyzed by overlaying LULC and the soil erosion maps in 2002 and 2015 (Table 11). This relationship is considered to be a valuable tool to monitor patterns of LULC change and the risk of soil erosion [9,104]. In comparison with the soil erosion based on the types of land use and land cover, the results revealed that human activities mainly influenced soil erosion concerning soil erosion risk, which was higher in rain-fed agricultural land and paddy field, highlighting their vulnerability to water-induced erosion, as compared to areas under forests (ever-green forest and deciduous forest), grassland, shrubland and built-up area. This can be explained by the intensive cultivation of crops in the Battambang province of Cambodia, the country's largest rice-producing province. As stated in the introduction [60] the area of rice production increased from 2.72 million ha in 2009 to 3.05 million ha in 2013. In the catchment, farmers practice conventional agricultural methods for crop production, leading to soil degradation. This tends to cause a higher rate of erosion and loss of soil organic matter content, which affects the stability of soil aggregate [15,105].

Forests, grassland and shrubland areas are also prone to soil erosion. However, due to better soil cover, the rate of erosion in these areas is lower than that of agricultural land and paddy fields. Table 11 shows that the forest lands under mixed forest experienced the highest soil erosion among the other types of land uses caused by the forest clearance by farmers to expand agricultural productivities. This observation aligns with a recent finding by Kong et al. [65], which demonstrated that from 2002 to 2010, forest conversion was relatively more intensive and homogenous in Pailin Province where the Stung Sangkae catchment covers almost one-third part of it (Figure 1). If the trend of transforming forestlands to agricultural lands continues to increase, the possibility of soil erosion will be further expanded, which will affect the sustainability of agricultural lands in the catchment for crop production. Therefore, agricultural deforestation must be minimized, especially on steep slopes. Additionally, it is necessary to implement agricultural management practices, such as on-farm conservation agriculture practices (CAP), water conservation and management, agroforestry practices, vegetation cover restoration and the creation of slopes terraces [15] to achieve sustainable control of soil erosion to improve the productivity of growing crops.

## 5. Conclusions

The map of LULC and findings clearly illustrate extensive soil erosion of very low to moderate severity rates ranging from 0.2 to 7.1 t/ha/y. The highest erosion rates of 14.3 to 62.9 t/ha/y were found in parts of the upland of the Stung Sangkae catchment, mainly due to steep slopes, high rate of erosion and degradation of the vegetation. Between 2002 and 2015, considerable changes in soil loss rate were observed in agricultural land. The forest lands decreased significantly during the investigated period, notably a massive shift in deciduous and mixed forest converted to agricultural land, paddy rice fields and other types of land use. Therefore, it is necessary to integrate protection measures at the farm level and target areas of high risk of erosion, mainly the degraded lands along the steep slopes, to limit the conversion of forest areas for agriculture and minimize the rate of erosion where the land is bare or with low vegetation cover. Some of the recommended measures to prevent soil erosion includes on-farm conservation agriculture practices (CAP), water conservation and management, agro-forestry practices, vegetation cover restoration and terracing. Future soil erosion assessment work in the study area should examine soil loss due to gully erosion, which is not currently possible using the RUSLE model. Additionally, calibration of the RUSLE results through field experiments helps to verify the accuracy of the estimated soil erosion in the study area.

**Author Contributions:** Conceptualization, N.N., M.M., J.J., G.S., M.R.R.; Methodology, N.N., M.M., J.J.; Software, N.N., J.J.; Formal analysis, N.N.; Investigation, M.M., G.S., M.R.R., B.N.; Resources, G.S., P.V.V.P., M.R.R.; Data curation, N.N.; Writing—original draft preparation, N.N.; Writing—review and editing, N.N., M.M., J.J., G.S., P.V.V.P., M.R.R.; Visualization, N.N.; Supervision, M.M.; J.J.; B.N.; Funding acquisition, G.S., M.R.R., M.M., P.V.V.P. All authors have read and agreed to the published version of the manuscript.

**Funding:** This research funded by the United States Agency for International Development (USAID) through Partnerships for Enhanced Engagement in Research (PEER) Cycle 7 implemented by the National Academy of Sciences (NAS) under sponsor (Cooperative Agreement No. AID-OAA-A-11-00012). This research is made possible by the generous support of the American People provided to the Center of Excellence on Sustainable Agricultural Intensification and Nutrition (CE SAIN) of the Royal University of Agriculture through the Feed the Future Innovation Lab for Collaborative Research on Sustainable Intensification at Kansas State University funded by the USAID under Cooperative Agreement No. AID-OAA-L-14-00006. The contents are the sole responsibility of the authors and do not necessarily reflect the views of USAID or the United States Government or any other organizations.

**Institutional Review Board Statement:** Not applicable.

**Informed Consent Statement:** Not applicable.

**Data Availability Statement:** Data is available upon request from the corresponding author.

**Acknowledgments:** The author deeply appreciates the valuable comments and suggestions in preparing this paper for publication from the editorial board of Sustainability and the two anonymous reviewers. Thanks to Kelly Robbins, Senior Program Officer, the National Academy of Sciences (NAS), for her support and guidance on the research grant implementation as well as acknowledge Partnerships for Enhanced Engagement in Research (PEER) for providing the research grant and the Center of Excellence on Sustainable Agricultural Intensification and Nutrition (CE SAIN), Royal University of Agriculture for providing the research scholarship. The author also thank the Ministry of Water Resources and Meteorology (MOWRAM) of Cambodia, Institute of Technology of Cambodia (ITC), and Mekong River Commission (MRC) for providing the data for the research. Contribution number 22-049-J from Kansas Agricultural Experiment Station.

**Conflicts of Interest:** The authors declare no conflict of interest.

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
