# Peer review of "Land Use and Land Cover Changes and Its Impact on Soil Erosion in Stung Sangkae Catchment of Cambodia"

_sustainability, doi:10.3390/su13169276_

Round 1
Reviewer 1 Report
I made some grammatical editing. Please, check your grammar and also try to use simpler sentence structure. I have made some recommendations (in red ink) and scanned and attached to this review.
Overall, this is a good piece of research.

Author Response
My co-authors and I are thankful for the edits of reviewer 1. We accepted all the edits and included them in the revised paper. Some examples are below, of the edits that we accepted and changed in the revised paper. The revised version was sent as a pdf file. We also included a REVISED VERSION in word that shows the edits we did using TRACK CHANGES. In the track changes version, you can see all the edits of reviewer 1 were incorporated.
Response to Reviewer 1 Comments
Line 18: The agricultural expansion
Response to line 18 is in line 18 of the revised version: “The agricultural expansion” is changed to “Agricultural expansion”
Line 19: a major environmental problems
Response to line 19 is in line 19 of the revised version: “a major environmental problems” is changed to “major environmental problems”
Line 21: This research was aimed to
Response to line 21 is in line 21 of the revised version: “This research was aimed to” is changed to “This research aimed to”
Line 39: “a threat to land” and “resulting in low agricultural productivity”
Response to line 39 is in line 39 and 40 of the revised version: “a threat to land” is changed to “poses a threat to land” and “resulting in low agricultural productivity” is changed to “This threat may result in low agricultural productivity”
Line 46: 10-15 times quicker than
Response to line 46 is in line 46 of the revised version: “10-15 times quicker than” is changed to “10-15 times higher than”
Line 48: can deteriorate turbidity
Response to line 48 is in line 48 of the revised version: “can deteriorate turbidity” is changed to “can increase turbidity”
Line 50: affect the people
Response to line 50 is in line 50 of the revised version: “affect the people” is changed to “affect people”
Line 57: the soil erosion
Response to line 57 is in line 57 of the revised version: “the soil erosion” is changed to “soil erosion”
Line 58: which were more challenging
Response to line 58 is in line 58 of the revised version: “which were more challenging” is changed to “These were more challenging”
Line 62: In the previous time
Response to line 62 is in line 62 of the revised version: “In the previous time” is changed to “In previous time”
Line 68: to analyze how
Response to line 68 is in line 69 of the revised version: “and to analyze how” is changed to “and analyzed how”
Line 75: Some of the recent studies
Response to line 75 is in line 75 of the revised version: “Some of the recent studies” is changed to “Some recent studies”
Line 83: the forest cover
Response to line 83 is in line 154 of the revised version: “the forest cover” is changed to “forest cover”
Line 86: Most of the soil erosion
Response to line 86 is in line 157 of the revised version: “Most of the soil erosion” is changed to “Most soil erosion ”
Line 88: due to the investments
Response to line 88 is in line 159 of the revised version: “due to the investments” is changed to “due to investments”
Line 97: According to Ministry of Environment
Response to line 97 is in line 169 of the revised version: “According to Ministry of Environment” is changed to “According to the Ministry of Environment”
Line 99: , primarily caused by
Response to line 99 is in line 171 of the revised version: “, primarily caused by” is changed to “This was primarily caused by”
Line 104: A recent finding by Lohani et al. [44] addressed that
Response to line 104 is in line 176 of the revised version: “A recent finding by Lohani et al. [44] addressed that” is changed to “A recent study by Lohani et al. [44] reported that”
Line 109: Tonle Sap region occurred relatively high
Response to line 109 is in line 181 of the revised version: “Tonle Sap region occurred relatively high” is changed to “Tonle Sap region was relatively high”
Line 124: The 65% of forest cover
Response to line 124 is in line 196 of the revised version: “The 65% of forest cover” is changed to “Sixty-five (65%) percent of the forest cover”
Line 152: Target 15.3 clearly state to combat
Response to line 152 is in line 226 of the revised version: “Target 15.3 clearly state to combat” is changed to “Target 15.3 clearly aims to combat”
Line 164: immediately or indirectly affords
Response to line 164 is in line 238 of the revised version: “immediately or indirectly affords” is changed to “directly or indirectly affords”
Line 168: the third-largest tributaries of the Tonle Sap
Response to line 168 is in line 242 of the revised version: “the third-largest tributaries of the Tonle Sap” is changed to “the third-largest tributary of the Tonle Sap”
Line 198: The FAO slope classification
Response to line 198 is in line 267 of the revised version: “The FAO slope classification” is changed to “FAO slope classification”
Line 201: the details are presented in Table 2 and Figure 3
Response to line 201 is in line 278 of the revised version: “the details are presented in Table 2 and Figure 3” is changed to “(see Table 2 and Figure 3)”
Line 217: The built-up areas was also increased
Response to line 217 is in line 296 of the revised version: “The built-up areas was also increased” is changed to “The built-up areas also increased”
Line 227: to estimate the soil erosion
Response to line 227 is in line 309 of the revised version: “to estimate the soil erosion rate” is changed to “to estimate soil erosion rate”
Line 259: especially in research areas
Response to line 259 is in line 344 of the revised version: “especially in research areas” is changed to “especially in the research areas”
Line 267: The equation has adopted by ..... and was extensively used in Thailand
Response to line 267 is in line 352-353 of the revised version: “The equation has adopted by ..... and was extensively used in Thailand” is changed to “The equation has been adopted by ..... and has been extensively used in Thailand”
Line 318: This equation is adopted by....
Response to line 318 is in line 404 of the revised version: “The equation is adopted by” is changed to “The equation has been adopted by”
Line 337: may differ between the two countries
Response to line 337 is in line 425 of the revised version: “may differ between the two countries” is changed to “may differ among different countries”
Line 355: by settling the value ... and as seen in the other studies
Response to line 355 is in line 442 of the revised version: “by settling the value” is changed to “by setting the value” and “as seen in the other studies” is changed to “as seen in other studies”
Line 399: result show that
Response to line 399 is in line 487 of the revised version: “result show that” is changed to “result shows that”
Line 433: soil loss rate were
Response to line 433 is in line 521 of the revised version: “soil loss rate were” is changed to “soil loss rates were”
Line 434: soil loss rate of slope
Response to line 434 is in line 522 of the revised version: “soil loss rate of slope” is changed to “soil loss rates of slope”
Line 444: The result also revealed that
Response to line 444 is in line 532 of the revised version: “The result also revealed that” is changed to “The results also revealed that”
Line 445: during the invested period
Response to line 445 is in line 533 of the revised version: “during the invested period” is changed to “during the investigated period”
...........
Reviewer 2 Report
This is an interesting approach regarding the soil erosion response to land use/cover changes. However, some serious changes are needed so as to improve the presentation of the results.
Indroduction
Add a comment about erosion models’ types (Empirical models, Physical-based models) differences and clarify the most well known.
A paragraph demonstrating the relationship between soil erosion and rainfall erosivity in terms of climate change should be added with appropriate references. This is important as climate change will increase and enchance the erodibility, especially in areas where land use change.
- Panagos, P., Ballabio, C., Meusburger, K., Spinoni, J., Alewell, C., & Borrelli, P. (2017). Towards estimates of future rainfall erosivity in Europe based on REDES and WorldClim datasets. Journal of Hydrology, 548, 251-262.
- Stefanidis, S., Alexandridis, V., Chatzichristaki, C., Stefanidis, P. (2021). Assessing Soil Loss by Water Erosion in a Typical Mediterranean Ecosystem of Northern Greece under Current and Future Rainfall Erosivity. Water, 13(15), 2002.
- Hateffard, F., Mohammed, S., Alsafadi, K., Enaruvbe, G. O., Heidari, A., Abdo, H. G., & Rodrigo-Comino, J. (2021). CMIP5 climate projections and RUSLE-based soil erosion assessment in the central part of Iran. Scientific reports, 11(1), 1-17.
Also, some comments that except future climate condition (climate change), observed trends in climate highly have affected soil erosion.
- Stefanidis, S., & Chatzichristaki, C. Response of soil erosion in a mountainous catchment to temperature and precipitation trends. Carpathian J. Earth Environ. Sci. 2017, 12, 35, 39.
- Bezak, N., Ballabio, C., Mikoš, M., Petan, S., Borrelli, P., & Panagos, P. (2020). Reconstruction of past rainfall erosivity and trend detection based on the REDES database and reanalysis rainfall. Journal of Hydrology, 590, 125372.
Clearly state the research gap, the innovative points and novelty of the current study and new aspect regarding the recent literature
Material and method
Add a flowchart of the methodology and a table summarizing the data used and data sources.
Results
It would be nice to add a map with spatial differences of soil loss due to land cover changes.
Author Response
My co-authors and I are thankful for the edits of reviewer 2. We have fulfilled and revised the manuscript based on the valuable comments from the reviewer 2 accordingly. Some paragraphs are added in the Point 1: Introduction, while the flowchart of the methodology and a table summarizing the data used and data sources are added in Point 2: Material and method. Moreover, a map with spatial differences of soil loss due to land cover changes and the table showing the distribution of soil erosion based on the LULC conversion are also added in the Point 3: Results. Please see the attachment.

Round 2
Reviewer 2 Report
All my comments has been addressed and the article in the current version can be accepted for publication